# The Mechanism of TT2-Type *MYB* Transcription Factor *JrMYB1L* in Anthocyanin Biosynthesis in ‘Jinghong 1’ Walnuts

**DOI:** 10.3390/plants14243727

**Published:** 2025-12-06

**Authors:** Suilin Zhang, Maofu Li, Wanmei Jin, Yunqi Zhang, Haigen Xu, Hanpin Li, Yonghao Chen, Zhixia Hou, Jianxun Qi

**Affiliations:** 1Beijing Engineering Research Center of Deciduous Fruit Tree, Institute of Forestry and Fruit Research, Beijing Academy of Agriculture and Forestry Sciences, Beijing 100093, China; lg20170404@bjfu.edu.cn (S.Z.);; 2State Key Laboratory of Efficient Production of Forest Resources, Beijing Forestry University, Beijing 100083, China; 3Key Laboratory for Silviculture and Conservation of Ministry of Education, Beijing Forestry University, Beijing 100083, China

**Keywords:** red walnut, Jinghong 1, anthocyanin biosynthesis, TT2-type *R2R3-MYB*, MBW complex

## Abstract

Red walnuts have been widely studied because of their strong antioxidant activity and ornamental value. However, research on the mechanism of anthocyanin biosynthesis in walnuts remains in the initial stage. The regulatory mechanism of TT2-type *R2R3-MYB* transcription factors in anthocyanin biosynthesis in walnuts is also unclear. Therefore, this study used ‘D2-1’ and ‘Jinghong 1’ walnuts as plant materials. The testa of ‘Jinghong 1’ was red, and its anthocyanin content was significantly higher than that of ‘D2-1’, mainly composed of cyanidin-3-O-arabinoside, cyanidin-3-O-galactoside, and cyanidin-3-O-glucoside. Differentially expressed genes between ‘D2-1’ and ‘Jinghong 1’ testa were enriched in phenylpropanoid biosynthesis and flavonoid biosynthesis. Next, this study identified a TT2-type *R2R3-MYB* transcription factor *JrMYB1L*, which was involved in regulating the anthocyanin biosynthesis in the testa of ‘Jinghong 1’. The overexpression of *JrMYB1L* could promote anthocyanin accumulation in walnut leaves and activate the expression of *JrCHS*, *JrCHI*, *JrF3H*, *JrDFR*, *JrANS*, *JrUFGT*, *JrLAR,* and *JrANR*. In addition, yeast two-hybrid results proved that JrMYB1L, JrbHLH42, and JrWD40 proteins could interact with each other. The results of yeast one-hybrid and dual-luciferase assays indicated that *JrMYB1L* could activate the expression of *JrCHS* and *JrUFGT* by binding to their promoters. Based on the above results, this study proposed a possible regulatory mechanism. *JrMYB1L* activated the expression of *JrCHS* and *JrUFGT* in the form of JrMYB1L-JrbHLH42-JrWD40 complex, thereby promoting anthocyanin accumulation in the testa of ‘Jinghong 1’. In summary, this study lays a theoretical foundation for revealing the regulatory mechanism of anthocyanin biosynthesis in red walnut and contributes to the breeding of new varieties of red walnuts with more edible and ornamental value.

## 1. Introduction

Walnuts (*Juglans regia* L.) are known as one of the four major nuts due to their high antioxidant content and rich nutrients such as vitamin E, folate, quercetin, melatonin, polyphenols, and linoleic acid [1]. ‘Robert Livermore’ was originally selected by crossing ‘Hoddle’ and ‘RA1088’, and it has the red testa and high anthocyanin content [2]. ‘Jinghong 1’, named after its red testa, is the offspring of ‘Robert Livermore’, which was selected by our research group in the previous period [1,3]. It has been recognized as a new variety by the National Forestry and Grassland Administration with the variety right number ‘20220628’. It has more stable characteristics, and its testa is red, which is consistent with ‘R2’ walnut [4]. ‘R2’ walnut, which also has the red testa, is preserved at the Jinniushan Base of Shandong Institute of Pomology (National Germplasm Repository of Walnut and Chestnut, Tai’an, Shandong, China) [4]. In recent years, red walnuts have broad market prospects and have become a characteristic industry for rural revitalization due to their attractive appearance, high nutrition, and high returns.

Anthocyanin originates from the phenylpropanoid biosynthesis pathway and are the secondary metabolites of flavonoids [5]. It is widely found in colorful plants such as sweet cherries [5,6], blueberries [7], and kiwifruit [8]. The flavonoid biosynthesis pathway is involved in regulating anthocyanin biosynthesis [4], procyanidins [9], and flavonols [10], and has been widely studied. This pathway contains many structural genes encoding anthocyanin biosynthesis enzymes, including *phenylalanine ammonia lyase (PAL)*, *cinnamate 4-hydroxylase (C4H)*, *4-coumarate: enzyme A ligase (4CL)*, *chalcone synthase (CHS)*, *chalcone isomerase (CHI)*, *flavanone 3-hydroxylase (F3H)*, *flavanone 3′-hydroxylase (F3′H)*, *dihydroflavanone 4-reductase (DFR)*, *anthocyanin synthase (ANS)*, and *UDP flavanone glucosyltransferase (UFGT)*. *Leucocyanidin reductase (LAR)* and *anthocyanin reductase (ANR)* are essential structural genes for procyanidin biosynthesis, while *flavonol synthase (FLS)* is a necessary structural gene for flavonol biosynthesis [5]. To some extent, the function of structural genes is highly conserved [11]. Anthocyanin biosynthesis is mainly regulated by transcription factors, such as *R2R3-MYB* [12,13,14]. The first *R2R3-MYB* transcription factor that activates anthocyanin biosynthesis was identified by Paz Ares et al. [15]. Subsequently, more *R2R3-MYB* transcription factors positively regulating anthocyanins were identified in tree species such as apples [16], blueberries [7], kiwifruit [8], pears [14], tulips [12], etc. For instance, bagging treatment upregulates the expression of *PavMYB10.1* and *PavNAC02,* thereby activating the expression of structural genes such as *PavANS* and *PavUFGT*, and promoting anthocyanin accumulation in sweet cherries [6]. Furthermore, the R2R3-MYB protein typically forms the MBW complex with bHLH and WD40 proteins, jointly regulating the expression of structural genes in the anthocyanin biosynthesis pathway [17,18,19]. For example, the *Arabidopsis* MBW complex regulates anthocyanin biosynthesis by regulating the expression of late structural genes *DFR*, *LDOX*, and *UFT75C1* [19].

The TT2-type *R2R3-MYB* transcription factor is usually considered to be involved in regulating proanthocyanin biosynthesis, such as in *Arabidopsis* [20], grapes [21], strawberries [22], apples [23], cocoa [24], and vanilla [25]. Interestingly, TT2-type *R2R3-MYB* transcription factors can positively regulate anthocyanin biosynthesis in peaches [26], kiwifruit [27], sweet gum [28], and *Chaenomeles speciosa* [29]. In *Prunus persica* cv. ‘Genpei’, *Peace* (*AtTT2/MYB123* homologous gene) was introduced into the petals through a gene gun, resulting in magenta spots [26]. In *Actinidia chinensis* cv. ‘Hongyang’, the co-expression of *AcMYB123* and *AcbHLH42* promotes anthocyanin synthesis by activating the expression of *AcF3GT1* and *AcANS* [27]. In *Liquidambar formosana*, *LfMYB123* promotes the accumulation of anthocyanins in young leaves by activating the expression of *LfF3′H1* and *LfDFR1* [28]. In *Chaenomeles speciosa*, the overexpression of *CsMYB123* activates the expression of *CsANS*, *CsCHI*, and *CsF3H*, causing fruits and leaves to turn red and increasing anthocyanin content. In addition, the silence of *CsMYB123* is the opposite [29]. This indicates that the TT2-type *R2R3-MYB* transcription factor has a certain broad-spectrum function and does not specifically regulate anthocyanin biosynthesis. In red walnut ‘RW-1’, the co-expression of *JrMYB12* (TT2-type *R2R3-MYB* transcription factor) and *JrbHLH42* promotes the accumulation of proanthocyanins by activating the expression of *JrLAR* and *JrANR*. However, *JrMYB12* did not have a direct effort on anthocyanin biosynthesis [9]. Therefore, further research is still warranted on the regulatory relationship between TT2-type *R2R3-MYB* transcription factors and anthocyanin biosynthesis.

At present, studies on the anthocyanin biosynthesis of walnuts mainly focus on transcription, metabolomics analysis, and identification of gene family members [30,31]. The difference between the exocarp of ‘Zhonglin1’ and ‘RW-1’ is mainly enriched in plant hormone signaling and phenylpropanoid biosynthesis, and the expression level of *JrUFGT5* is significantly positively correlated with anthocyanin content [30]. In the hybrid offspring of ‘Zhonglin1’ and ‘RW-1’, the differential genes between the red leaf offspring and the green leaf offspring are mainly enriched in phenylpropanoid biosynthesis and flavonoid biosynthesis [31]. The expression levels of *JrEGL1a*, *JrEGL1b*, *JrbHLHA1,* and *JrbHLHA2* in the leaves of red leaf offspring reached their peak during the SR-1 period (the full red period of red leaves in seeding production), which may be the reason for the accumulation of anthocyanins in the leaves [11]. The expression of *JrWD40-133*, *JrWD40-150*, *JrWD40-155*, and *JrWD40-206* in the exocarp of ‘Zijing’ was significantly higher than that in ‘Lvling’, which may promote anthocyanin synthesis. However, the expression of *JrWD40-65*, *JrWD40-172*, *JrWD40-191*, *JrWD40-224*, and *JrWD40-254* in the exocarp of ‘Zijing’ was significantly lower than that in ‘Lvling’, which may inhibit anthocyanin synthesis [32]. The expression of *JrMYB22*, *JrMYB23*, *JrMYB24*, *JrMYB27*, *JrMYB115*, *JrMYB129*, *JrMYB194*, *JrMYB198*, and *JrMYB217* in ‘Zijing’ leaves was significantly higher than that in ‘Lvling’, which may positively regulate the anthocyanin synthesis in walnut purple red leaves [33]. In red walnuts ‘R1’ and ‘R2’, *JrMYB113* promotes the expression of *JrLDOX-3* and *JrUAGT-3*, while *JrMYB27* promotes the accumulation of anthocyanins by promoting the expression of *JrLDOX-2*. In the exocarp of ‘R2’, the silence of *JrATHB-12* can increase the specific expression of *JrMYB113* [4]. In addition, the insertion of the methylated transposon Gypsy TE of *MIEL1* inhibits its expression, reduces anthocyanin accumulation, and results in a yellow instead of red stigma of walnut female flowers [34]. Overall, further exploration is needed to study the regulatory mechanisms of walnut anthocyanin biosynthesis. However, the relationship and regulatory mechanism between TT2-type *R2R3-MYB* transcription factors and walnut anthocyanin biosynthesis are still unclear.

Therefore, in this study, ordinary walnut ‘D2-1’ and red walnut ‘Jinghong 1’ were used as test materials, and a TT2-type *R2R3-MYB* transcription factor (*JrMYB1L*) was screened based on transcriptome and metabolome analysis. The presence of the JrMYB1L-JrbHLH42-JrWD40 complex was preliminarily confirmed through yeast two-hybrid experiments. Studied the binding of *JrMYB1L* transcription factor to *JrCHS* and *JrUFGT* promoters using yeast one-hybrid and dual luciferase assays, and investigated the roles of *JrMYB1L* in anthocyanin biosynthesis through transient transformation. In summary, this study provides a scientific basis for improving the regulatory network of anthocyanin biosynthesis in ‘Jinghong 1’ and lays a theoretical foundation for breeding walnuts with special colors.

## 2. Results

### 2.1. The Accumulation of Anthocyanins During Walnut Development

According to the color changes in the walnut testa of ‘Jinghong 1’, the developmental stages of walnuts were divided into green fruit, testa color transition, and ripening stages (Figure 1a–c). Throughout the entire developmental period, ‘D2-1’ and ‘Jinghong 1’ both had green exocarp, but showed significant differences in testa color. In the green fruit stage, the kernel and testa of ‘D2-1’ and ‘Jinghong 1’ were both milky white (Figure 1a). In the testa color transition stage, the kernel and testa of ‘D2-1’ and the kernel of ‘Jinghong 1’ were both milky white. But the testa of ‘Jinghong 1’ turned to light red (Figure 1b). In the ripening stage, the kernel of ‘D2-1’ and ‘Jinghong 1’ was both milky white. However, the testa of ‘D2-1’ has turned to light yellow, and the testa of ‘Jinghong 1’ has turned to dark red (Figure 1c). In addition, young leaves of ‘D2-1’ appeared green, while the tips of young leaves of ‘Jinghong 1’ appeared red and the base was light green (Figure 1d). The anthocyanin content in young leaves of ‘Jinghong 1’ was also significantly higher than that of ‘D2-1’ young leaves. In the ripening stage, the anthocyanin content of the ‘Jinghong 1’ testa was significantly higher than that of the ‘D2-1’ testa (Figure 1e,f). These results all demonstrated the accumulation of anthocyanins in the young leaves and testa of ‘Jinghong 1’.

To further investigate the differences in flavonoid metabolites between the fruits of ‘D2-1’ and ‘Jinghong 1’, the testa and kernel of ‘D2-1’ and ‘Jinghong 1’ in three stages were selected for ultra-high performance liquid chromatography-mass spectrometry (UPLC-MS/MS) analysis. A total of 85 anthocyanins, 8 flavonoids, and 6 anthocyanins were discovered. Anthocyanins include 16 derivatives of cyanidins, 16 derivatives of paeoniflorin, 15 derivatives of delphinidin, 15 derivatives of geraniums, 12 derivatives of mallow pigments, and 11 derivatives of petunians (Appendix A). The differential metabolites between the testa of ‘D2-1’ and ‘Jinghong 1’ were mainly cyanidin-3-O-arabinoside, cyanidin-3-O-galactoside, cyanidin-3-O-glucoside, and cyanidin-3-O-xyloside (Figure 2a, Appendix A). These substances were significantly upregulated in the testa of ‘Jinghong 1’. Among them, the content of cyanidin-3-O-galactoside was the highest, accounting for 45.10%, 48.34%, and 37.15% in the green fruit, testa color transition, and ripening stages, respectively (Appendix A). The difference in cyanidin-3-O-arabinoside was the most significant, with its content in the ‘Jinghong 1’ testa being 7318.32 times that of the ‘D2-1’ testa. Flavonoids included arbutin, kaempferol-3-O-rutinoside, dihydroquercetin, naringenin (dihydrokaempferol), naringenin-7-O-glucoside, quercetin-3-O-glucoside (isoquercetin), rutin, and naringenin. Proanthocyanins included proanthocyanins A1, A2, B1, B2, B3, and C1. Among them, the differences in kaempferol-3-O-rutinoside and proanthocyanin A1 were the most significant. In the ripening stage, the content of kaempferol-3-O-rutinoside and proanthocyanin A1 was 65.51 and 15.58 µg/g in the ‘Jinghong 1’ testa, but they were not detected in the ‘D2-1’ testa (Appendix A). Except for cyanidin-3-O-galactoside, there were almost no other anthocyanins in walnut kernels (Figure 2b, Appendix A). In summary, it was speculated that cyanidin-3-O-galactoside and cyanidin-3-O-arabinoside might be the main metabolites involved in the accumulation of anthocyanins in the ‘Jinghong 1’ red testa.

### 2.2. Differentially Expressed Genes (DEGs) During Walnut Development

To further clarify the mechanism of testa color formation between ‘D2-1’ and ‘Jinghong 1’, transcriptome sequencing was performed on walnut testa in the green fruit, testa color transition, and ripening stages. Using the ‘D2-1’ testa as the control group, the expression of genes in ‘Jinghong 1’ was analyzed (Appendix A). In the green fruit stage, there were a total of 2441 DEGs, with 1100 upregulated genes and 1341 downregulated genes. In the testa color transition stage, there were a total of 2380 DEGs, with 1098 upregulated genes and 1282 downregulated genes. In the ripening stage, there were a total of 2071 DEGs, with 1192 upregulated genes and 879 downregulated genes.

Performed KEGG enrichment analysis on DEGs between the testa of ‘D2-1’ and ‘Jinghong 1’. In the green fruit stage, DEGs were mainly enriched in plant–pathogen interactions, endoplasmic reticulum protein processing, and sulfur metabolism (Figure 2c). In the testa color transition stage, DEGs were mainly enriched in plant–pathogen interactions, biosynthesis of secondary metabolites, phenylpropanoid biosynthesis, starch and sucrose metabolism, and sulfur metabolism (Figure 2d). In the ripening stage, DEGs were mainly enriched in the biosynthesis of secondary metabolites, sulfur metabolism, phenylpropanoid biosynthesis, plant–pathogen interactions, biosynthesis of various plant secondary metabolites, and flavonoid biosynthesis (Figure 2e).

Time series analysis was performed on DEGs, resulting in a total of three clusters (Figure 3a–c). Cluster 1 consisted of 8673 genes, with low expression in the green fruit stage, the expression gradually increased and reached its peak in the ripening stage (Figure 3a). This might positively regulate the anthocyanin biosynthesis in the ‘Jinghong 1’ testa. Cluster 2 consisted of 9023 genes, and the expression showed a trend of first increasing and then sharply decreasing. In addition, the expression of ‘Jinghong 1’ testa was higher than that of ‘D2-1’ (Figure 3b). Cluster 3 consisted of 8145 genes, with the highest expression in the green fruit stage. As the fruit ripened, its expression gradually decreased (Figure 3c). It might negatively regulate the anthocyanin biosynthesis process. Clusters 1, 2, and 3 were all mainly enriched in plant–pathogen interaction, ribosome, and plant hormone signal transduction (Appendix A). Differently, cluster 1 was also enriched in phenylpropanoid biosynthesis (Appendix A). Therefore, it was believed that cluster 1 was the core module involved in anthocyanin biosynthesis.

Further WGCNA analysis was conducted, and a total of 17 co-expressed modules were identified (Figure 3d). Through module-anthocyanin content correlation analysis, it was found that three modules were significantly positively correlated with the content of cyanidin-3-O-galactoside and cyanidin-3-O-arabinoside, including sienna 3, darkgrey, and yellow green modules (Figure 3e). Among them, the darkgrey module was significantly positively correlated with both cyanidin-3-O-galactoside and cyanidin-3-O-arabinoside content (R^2^ < 0.01). Therefore, it was believed that the darkgrey module played important roles in the biosynthesis of anthocyanins in the ‘Jinghong 1’ testa. KEGG enrichment was performed on the darkgrey module, and it was mainly enriched in plant–pathogen interaction, plant hormone signal transduction, protein processing in endoplasmic reticulum, glutathione metabolism, flavonoid biosynthesis, and ABC transporters (Appendix A). Based on the KEGG enrichment (Figure 2c–e), time series analysis (Figure 3a–c), and WGCNA (Figure 3d,e) analysis results of DEGs, it was believed that phenylpropanoid biosynthesis and flavonoid biosynthesis were the main reasons for anthocyanin biosynthesis in the ‘Jinghong 1’ red testa.

### 2.3. Structural Genes of the Anthocyanin Biosynthesis Pathway and Expression Characteristics of JrMYB Transcription Factors

Phenylpropanoid biosynthesis is the starting point of anthocyanin biosynthesis, and flavonoid biosynthesis is the core pathway of anthocyanin biosynthesis [5]. In the flavonoid biosynthesis pathway, the structural genes *JrPAL*, *JrC4H*, *Jr4CL* (early stage), *JrCHS*, *JrCHI*, *JrF3H*, *JrF3′H* (mid stage), *JrDFR*, *JrANS*, and *JrUFGT* (late stage) jointly regulated anthocyanin biosynthesis. To further elucidate the regulatory mode of anthocyanin accumulation in the ‘Jinghong 1’ red testa, the expression levels of structural genes in this pathway were analyzed (Figure 4a). In the green fruit stage, *JrPAL*, *Jr4CL*, *JrCHS*, and *JrUFGT* all showed the highest number of significantly upregulated genes, with three genes each. Conversely, *JrPAL* showed the highest number of significantly downregulated genes, a total of four, followed by *Jr4CL* and *JrDFR*, with three each. In the testa color transition stage, *JrCHS* had the highest number of significantly upregulated genes, at five. While *JrPAL* had the highest number of significantly downregulated genes, at five. In the ripening stage, *JrPAL* had the highest number of significantly upregulated genes, reaching seven. *Jr4CL* and *JrUFGT* had the highest number of significantly downregulated genes, with two each. With the development of fruits, the expression level of *JrPAL* (*LOC108988497*, *LOC109008959*, *LOC118349603*), *JrC4H* (*LOC108996947*, *LOC109002391*), *Jr4CL* (*LOC108998277*), *JrCHS* (*LOC108988452*), *JrCHI* (*LOC108996546*, *LOC109007666*), *JrF3′H* (*LOC109011206*, *LOC109011249*, *LOC109017724*), *JrDFR* (*LOC108991381*, *LOC109010696*, *LOC109010697*), and *JrUFGT* (*LOC108998553*, *LOC108998554*) showed the upward trend. Based on the above results, it was found that *JrPAL* (*LOC109008959*), *JrC4H* (*LOC108996947*), *Jr4CL* (*LOC108998277*), *JrCHS* (*LOC108984452*), *JrCHI* (*LOC109007666*), *JrF3H* (*LOC108997708*), *JrF3′H* (*LOC109017724*), *JrDFR* (*LOC108991381*), *JrANS* (*LOC109010746*), and *JrUFGT* (*LOC109004490*) played important roles in the formation of red testa in ‘Jinghong 1’. qRT-PCR was used to validate *JrCHS*, *JrCHI*, *JrF3H*, *JrDFR*, *JrANS*, and *JrUFGT* (Figure 5), and the results showed that the qRT-PCR results were consistent with the transcriptome results, indicating that the transcriptome results were reliable.

To clarify the transcription factors that regulate the structural genes of the anthocyanin biosynthesis pathway, the differential expression characteristics of transcription factors were analyzed (Figure 4b). *JrMYB* has the highest number of transcription factors, with a total of 142. In the green fruit stage, 44 *JrMYBs* were significantly upregulated, and 22 *JrMYBs* were significantly downregulated. In the testa color transition stage, 41 *JrMYBs* were significantly upregulated, and 23 *JrMYBs* were significantly downregulated. In the ripening stage, 60 *JrMYBs* were significantly upregulated and 18 *JrMYBs* were significantly downregulated. 27 *JrMYBs* showed the upward trend with fruit development. In addition, the transcription levels of 12 *JrMYBs* in the testa color transition stage were significantly higher than those in the green fruit stage, and the transcription levels of 20 *JrMYBs* in the ripening stage were significantly higher than those in the testa color transition stage. Among them, 10 *JrMYBs* were significantly upregulated in the green fruit, testa color transition, and ripening stages, namely *LOC109001985*, *LOC109001129*, *LOC108995419*, *LOC109021106*, *LOC109011978*, *LOC108996040*, *LOC108988219*, *LOC109009038*, *LOC108994722*, and *LOC108984049*. During the transition of the testa of ‘Jinghong 1’ from the green fruit stage to the testa color transition stage, *LOC109001985* showed the most significant differences (Appendix A). It was speculated that it played an important role in the anthocyanin biosynthesis in the testa of ‘Jinghong 1’.

### 2.4. Combined Analysis of Transcriptome and Metabolome

Based on a pearson correlation coefficient ≥ 0.80 and *p*-value ≤ 0.05, correlation expression trend analysis was performed on differentially expressed genes and metabolites (Appendix A). Using cyanidin-3-O-arabinoside (identified by screening in Section 2.1) as the key metabolite, DEGs between the testa of ‘D2-1’ and ‘Jinghong 1’ in the testa color transition and ripening stages were analyzed. Based on the integration of correlation analysis results, 2 *JrMYBs* closely related to cyanidin-3-O-arabinoside were screened, namely *LOC109007798* and *LOC109001985*. Xu et al. [4] have confirmed that *LOC109007798* can positively regulate anthocyanin synthesis of ‘R2’ testa. Based on the results of the analysis in Section 2.3, this study mainly focused on *LOC109001985.*

### 2.5. Multiple Sequence Alignment, Phylogenetic Analysis, and Tissue-Specific Expression of JrMYB1L

According to the annotation results of *LOC109001985* in the NR database, *LOC109001985* has been annotated as the transcription factor *MYB1-like*. Therefore, the gene was named *JrMYB1L*. The prediction of the conserved domain of the JrMYB1L protein revealed the presence of the PLN03212 domain at its N-terminus (Figure 6a). This indicated that the JrMYB1L protein belonged to R2R3-MYB proteins. Multiple sequence alignment was performed between the JrMYB1L protein sequence and the TT2-type R2R3-MYB protein sequences of *Arabidopsis*, *Chaenomeles speciosa*, and peach. The JrMYB1L protein exhibited the typical R2R3 repeat at the N-terminus and the [V/A] [I/V] RTKA [T/I/A/L] [R/K] motif (TT2-domain) at the C-terminus (Figure 6b). This indicated that the JrMYB1L protein belonged to the typical TT2-type R2R3-MYB proteins. Systematic evolutionary analysis was conducted on JrMYB1L and R2R3-MYB proteins from different species. JrMYB1L showed the closer evolutionary relationship with CiMYB1-like (American walnut) and CaMYB1-like (European hazelnut) proteins, while the relatively distant evolutionary relationship with AcMYB123 (kiwifruit) protein (Figure 6c).

The expression levels of *JrMYB1L*, *JrCHS,* and *JrUFGT* were significantly positively correlated with the content of cyanidin-3-O-galactoside and cyanidin-3-O-arabinoside selected in Section 2.1 (Appendix A). To verify the roles of *JrMYB1L*, *JrCHS*, and *JrUFGT* in anthocyanin biosynthesis, the expression characteristics of ‘D2-1’ and ‘Jinghong 1’ leaves, stems, testa, and kernel were analyzed separately (Figure 6d–f). In ‘D2-1’, *JrMYB1L* had the highest expression level in the stems, followed by the kernel and leaves, and the lowest expression level in the testa. *JrCHS* and *JrUFGT* had the highest expression levels in stems, followed by leaves and testa, and the lowest expression levels in the kernel. In ‘Jinghong 1’, the expression level of *JrMYB1L* was significantly higher in the kernel than in other parts, followed by the leaves and testa, with the lowest expression level in the stems. *JrCHS* had the highest expression level in the testa, followed by the kernel and stems, and the lowest expression level in the leaves. *JrUFGT* had the highest expression level in leaves, followed by stems and kernel, and the lowest expression level in testa. In addition, during the fruit development of ‘Jinghong 1’, *JrMYB1L, JrCHS,* and *JrUFGT* expression levels of the testa in the ripening stage were significantly higher than those in the green fruit stage (Figure 6g). This indicated that *JrMYB1L, JrCHS,* and *JrUFGT* all contributed to anthocyanin biosynthesis.

### 2.6. The Role of JrMYB1L in Anthocyanin Biosynthesis of Walnuts

In the green fruit stage, *JrMYB1L* expression level in the testa of ‘D2-1’ and ‘Jinghong 1’ was extremely low. In the testa color transition and ripening stages, *JrMYB1L* expression level in the testa of ‘Jinghong 1’ was significantly higher than that of ‘D2-1’. *JrMYB1L* expression level in the testa of ‘Jinghong 1’ showed the upward trend with fruit development (Figure 4b and Figure 6d). To elucidate the role of *JrMYB1L* in anthocyanin biosynthesis, the transient overexpression of *JrMYB1L* was conducted in walnut leaves. In walnut seedlings overexpressing *JrMYB1L*, point distribution of anthocyanins was observed in leaves (Figure 7a and Appendix A). The expression level of *JrMYB1L* was also significantly higher than that of the control group (Figure 7b). This indicated that the method of instantaneous conversion was feasible and the effect was significant. To further clarify the role of *JrMYB1L* in walnut anthocyanin biosynthesis, the expression characteristics of genes related to anthocyanin and proanthocyanin biosynthesis pathways were analyzed separately. In walnut seedlings overexpressing *JrMYB1L*, the expression levels of *JrCHS*, *JrCHI*, *JrF3H*, *JrDFR*, *JrANS*, *JrUFGT*, *JrLAR*, and *JrANR* were significantly higher than those in the control group (Figure 7b). This indicated that *JrMYB1L* might activate anthocyanin and proanthocyanin biosynthesis.

### 2.7. The Interaction Relationship Between JrMYB1L, JrbHLH42, and JrWD40 Proteins

To investigate the physical interactions between JrMYB1L, JrbHLH42, and JrWD40 proteins, Y2H experiments were conducted. Using pGBKT7 + pGADT7 as the negative control, pGBKT7-JrbHLH42, pGBKT7-JrWD40, and pGADT7 were co-transferred into Y2HGold cells. PGBKT7 + pGADT7, pGBKT7-JrbHLH42 + pGADT7, and pGBKT7-JrWD40 + pGADT7 were grown on SD/-Leu/-Trp plates, but did not grow on SD/-His/-Leu/-Trp and SD/-Ade/-His/-Leu/-Trp plates (Figure 8a). This indicated that JrbHLH42 and JrWD40 proteins do not have self-activating activity. pGBKT7-JrbHLH42 + pGADT7-JrMYB1L, pGBKT7-JrWD40 + pGADT7-JrMYB1L, and pGBKT7-JrWD40 + pGADT7-JrbHLH42 were grown on SD/-Leu/-Trp and SD/-His/-Leu/-Trp plates, but did not grow on SD/-Ade/-His/-Leu/-Trp plates (Figure 8b). This indicated that the above combination activated the transcription of His reporter genes, but could not activate the transcription of Ade reporter genes. In summary, JrMYB1L, JrbHLH42, and JrbHLH42 proteins could interact with each other.

### 2.8. JrMYB1L Positively Regulates the Transcription of JrCHS and JrUFGT

To elucidate the regulatory relationship between *JrMYB1L* and structural genes *JrCHS* and *JrUFGT*, the correlation analysis was conducted. We found that there was a significant positive correlation between the expression levels of *JrMYB1L* and the intermediate structural gene *JrCHS* and late structural gene *JrUFGT* in anthocyanin biosynthesis (Figure 8c). Furthermore, both *JrCHSpro* and *JrUFGTpro* have *MYB* binding sites (Figure 8d). To clarify the regulatory relationship between *JrMYB1L* transcription factor and *JrCHSpro* and *JrUFGTpro*, Y1H (Figure 8e) and dual luciferase assays (Figure 8f) were conducted separately. Using pGADT7 + pAbAi as the negative control, pAbAi-*JrCHSpro1427-1509* + pGADT7 and pAbAi-*JrUFGTpro1200-1400* + pGADT7 were grown on SD/-Ura plates (Appendix A). This indicated that *JrCHSpro1427-1509* and *JrUFGTpro1200-1400* had certain self-activating activity. After adding 950 ng/mL AbA, yeast cell growth was inhibited to some extent (Appendix A). This indicated that 950 ng/mL AbA could inhibit the self-activation activity of *JrCHSpro1427-1509* and *JrUFGTpro1200-1400*. pAbAi-*JrCHSpro1427-1509* + pGADT7 and pAbAi-*JrUFGTpro1200-1400* + pGADT7 could grow on both SD/-Leu and SD/-Leu + 950ng/mL AbA plates (Figure 8e). This indicated that *JrMYB1L* could bind to the promoters of *JrCHS* and *JrUFGT*.

SK + LUC-*JrCHSpro1427-1509* and SK + LUC-*JrUFGTpro1200-1400* were used as control groups, while SK-*JrMYB1L* + LUC-*JrCHSpro1427-1509* and SK-*JrMYB1L* + LUC-*JrUFGTpro1200-1400* were used as experimental groups. It was found that the fluorescence signal of the experimental group was significantly stronger than that of the control group (Figure 8f). In addition, the LUC/REN ratio of SK-*JrMYB1L* + LUC-*JrCHSpro1427-1509* was 2.68 times that of SK + LUC-*JrCHSpro1427-1509*, and the LUC/REN ratio of SK-*JrMYB1L* + LUC-*JrUFGTpro1200-1400* was 1.25 times that of SK + LUC-*JrUFGTpro1200-1400* (Figure 8f). This indicated that *JrMYB1L* could activate the promoters of *JrCHS* and *JrUFGT*. In summary, both Y1H and dual luciferase assays have demonstrated that *JrMYB1L* could positively regulate the transcription of *JrCHS* and *JrUFGT*.

## 3. Discussion

As an important secondary metabolite, anthocyanins have been widely studied by scientific researchers due to their antioxidant properties and ornamental value [4,35]. At present, red walnuts have great market prospects due to their high esthetic value and traditional red auspicious meaning. The anthocyanin levels in the young leaves and testa of ‘Jinghong 1’ were significantly higher than those of ‘D2-1’, with anthocyanin content in the testa around 1.4 mg/kg (Figure 1e,f). The abundant presence of cyanidin-3-O-arabinoside, cyanidin-3-O-galactoside, and cyanidin-3-O-glucoside resulted in the bright red color on the testa of ‘Jinghong 1’ (Figure 2a, Appendix A). This was consistent with the anthocyanin components in the ‘R2’ testa [4]. In Li et al.’s study, only the red walnut testa in the ripening stage was detected to contain delphinidin-3-O-glucoside, which was only 0.283 mg/kg [36]. This indicated that the types and contents of anthocyanins present in walnut testa were closely related to the variety. The anthocyanin content in the testa of ‘Jinghong 1’ in this study was significantly higher than that of the red walnut mentioned by Li et al. [36], which had certain advantages. It was worth noting that ‘Jinghong 1’ has been approved as a new variety by the National Forestry and Grassland Administration, with more stable traits and excellent specificity, consistency, and stability. In addition, the exocarp of some walnut varieties also appeared red, with strong ornamental value, such as ‘R1’ and ‘RW-1’. The exocarp of anthocyanin in ‘R1’ was mainly composed of cyanidin-3-O-galactoside, with the content of about 32.235 µg/g [4]. The content of anthocyanins in the exocarp of ‘RW-1’ increased with fruit development, ranging from 0.40 to 0.80 mg/kg [36]. However, the anthocyanin content in the testa of ‘R1’ was significantly lower than that of ‘R2’ [4]. It might be that the high anthocyanin levels in the exocarp of ‘R1’ affected the anthocyanin content in its testa. In the future, the breeding direction of high-quality walnut varieties will shift towards cultivating walnut varieties with high levels of anthocyanins in both exocarp and testa. This aims to enhance both its ornamental and edible value.

At present, the mechanism of anthocyanin biosynthesis in red walnuts is still in its infancy, mainly through the joint analysis of transcriptome and metabolome [4,30,36]. This study preliminarily revealed the regulatory mechanism of anthocyanin differences between the testa of ‘D2-1’ and ‘Jinghong 1’ through combined transcriptome and metabolome analysis, laying a theoretical foundation for further in-depth research. The enrichment pathways of DEGs between ‘D2-1’ and ‘Jinghong 1’ were mainly involved in phenylpropanoid biosynthesis and flavonoid biosynthesis (Figure 2c–e and Figure 3). This was consistent with the transcriptome results between ‘Zhonglin1’ and ‘RW-1’ [30]. The DEGs between green leaves and red leaves in the hybrid offspring of ‘RW-1’ were mainly enriched in metabolism, secondary metabolite biosynthesis, phenylpropanoid biosynthesis, and flavonoid biosynthesis [31]. This was highly consistent with the results of differential gene analysis of the testa. This indicated that the mechanism of anthocyanin biosynthesis was not closely related to plant tissues. During the fruit development of ‘Jinghong 1’, the expression levels of *JrPAL*, *JrC4H*, *Jr4CL*, *JrCHS*, *JrCHI*, *JrF3H*, *JrF3′H*, *JrDFR*, *JrANS*, and *JrUFGT* in the testa showed the upward trend (Figure 4a). Meanwhile, the expression levels of these genes in the ‘Jinghong 1’ testa were significantly higher than those in the ‘D2-1’ testa (Figure 4a). All of these results indicated the important role of phenylpropanoid and flavonoid biosynthesis pathways in the biosynthesis of anthocyanins in the testa of ‘Jinghong 1’. In addition, the enrichment pathway of ‘D2-1’ and ‘Jinghong 1’ testa DEGs involved plant–pathogen interactions (Figure 2c–e). This was consistent with the comparative analysis between the testa of ordinary walnuts and red-fleshed walnuts [36]. This indicated that anthocyanins were closely related to the disease resistance of plants. Anthocyanins can maintain cellular homeostasis and enhance plant resistance to pathogens by clearing the reactive oxygen species produced by pathogen invasion [37]. Therefore, future research can focus on the regulatory mechanisms of anthocyanin involvement in plant disease resistance.

The *JrMYB* transcription factor had the highest number of differential transcription factors between the testa of ‘D2-1’ and ‘Jinghong 1’ (Figure 4b). There were a total of 204 *JrR2R3-MYB* transcription factors in walnuts [33]. This study combined the expression characteristics of *JrR2R3-MYB* transcription factors and the differential metabolite gene correlation network to screen 2 *JrR2R3-MYB* transcription factors, namely *LOC109007798 (JrMYB113)* and *LOC109001985 (JrMYB1L)* (Figure 4b, Appendix A). *JrMYB113* transient overexpression could promote the accumulation of anthocyanins in the exocarp of walnuts [4]. Differently, *JrMYB1L* possesses the TT2 domain (Figure 6b) and belongs to the TT2-type *R2R3-MYB* transcription factor. In Zuo et al.’s study, *JrMYB1L* was named *JrMYB100* [33]. However, *JrMYB113* is a homologous gene of *AtMYB114* and does not belong to the TT2-type *R2R3-MYB* transcription factor [4]. Usually, TT2-type *R2R3-MYB* transcription factors are involved in proanthocyanin biosynthesis [23,24,25]. In ‘RW-1’, the co-expression of *JrMYB12* and *JrbHLH42* could promote anthocyanin biosynthesis [9]. In this study, although the transient overexpression of *JrMYB1L* resulted in a small amount of red dots on walnut leaves (Figure 7a), there was no significant accumulation of anthocyanins. This indicated that *JrMYB1L* could promote anthocyanin biosynthesis to some extent. The transient overexpression of *JrMYB1L* promoted the expression of *JrCHS*, *JrCHI*, *JrF3H*, *JrDFR*, *JrANS*, *JrUFGT*, *JrLAR*, and *JrANR* (Figure 7b). This indicated that *JrMYB1L* could activate the expression of the aforementioned genes and regulate anthocyanin and proanthocyanin biosynthesis. In peach [26], red-centered kiwifruit [27], and *Chaenomeles speciosa* [29], TT2-type *R2R3-MYB* transcription factors were involved in regulating anthocyanin biosynthesis. This was consistent with the results of this study. This study lays a theoretical foundation for the in-depth study of the relationship between TT2-type *R2R3-MYB* transcription factors and anthocyanin biosynthesis, which helps to enrich the regulatory network of walnut anthocyanin biosynthesis. Unfortunately, no significant anthocyanin was observed in walnut leaves overexpressing *JrMYB1L*. This might be due to the thick cuticle layer and complex vascular system distribution of walnut leaves, which prevented bacterial fluid from expanding in the leaves. It might also be related to the abundant presence of phenolic substances in walnut leaves, affecting the color of anthocyanins.

MYB-bHLH-WD40 participates in the biosynthesis of anthocyanins in the form of the complex [38]. In walnuts, JrMYB1L, JrbHLH42, and JrWD40 proteins could interact with each other and form the JrMYB1L-JrbHLH42-JrWD40 complex (Figure 8b). In the study of Zhao et al., JrMYB12 protein can interact with JrbHLH42 protein, which was consistent with the results of this study [9]. These all supported the results of this study. Furthermore, both Y1H and dual luciferase assays demonstrated that *JrMYB1L* could bind to the promoter regions of *JrCHS* and *JrUFGT* (Figure 8e,f). This was consistent with the positive correlation between the expression levels of *JrMYB1L, JrCHS,* and *JrUFGT* (Figure 8c). In addition, transient overexpression of *JrMYB1L* significantly increased the expression levels of *JrCHS* and *JrUFGT* (Figure 7b). The highest expression level of *JrCHS* was observed in the ‘Jinghong 1’ testa in the ripening stage (Figure 6g), indicating that *JrCHS* played a crucial role in its anthocyanin synthesis process. *JrMYB* transcription factor could activate the transcription of other downstream genes. For example, *JrMYB27* promoted anthocyanin synthesis of ‘R1’ exocarp and testa by activating the expression of *JrLDOX-2*. *JrMYB113* promoted anthocyanin synthesis of ‘R2’ testa by activating the expression of *JrLDOX-2* and *JrUAGT-3* [4]. Furthermore, *JrUC3GalT* was highly expressed in the testa of ‘Robert Livermore’. However, overexpression of *JrUC3GalT* in ‘Chandler’ (light colored testa) somatic embryos did not show a significant phenotype, which might be related to insufficient anthocyanin or UDP-galactose content in the endosperm [39]. This further indicated that anthocyanin biosynthesis was subject to complex regulation by multiple factors. The absence of any signal might lead to the inability of anthocyanin accumulate anthocyanins.

In summary, this study proposed a regulatory mechanism for anthocyanin biosynthesis in the testa of ‘Jinghong 1’. *JrMYB1L* activated the expression of downstream structural genes *JrCHS* and *JrUFGT* in the form of JrMYB1L-JrbHLH42-JrWD40 complex, thereby promoting anthocyanin biosynthesis in the testa of ‘Jinghong 1’ (Figure 9). However, what signal stimulated the extensive expression of *JrMYB1L* in the testa of ‘Jinghong 1’? Was it related to *JrMYB113*? Therefore, by predicting the relationship between JrMYB1L and JrMYB113 proteins, it was found that there might be an interaction between JrMYB1L and JrMYB113 proteins (Appendix A). However, further in-depth research and exploration are still needed regarding the relationship between the two and their regulatory mechanisms.

## 4. Materials and Methods

### 4.1. Plant Materials

The plant materials for this study, ordinary walnut ‘D2-1’ and red walnut ‘Jinghong 1’ (variety right number ‘20220628’). Their fruits were collected on 25 June, 2 August, and 12 August, respectively. In addition, they were both collected from the Walnut Base of the Forestry Fruit Tree Research Institute of the Beijing Academy of Agriculture and Forestry Sciences, located in Wanghu Village, Dasungezhuang Town, Shunyi District, Beijing. Slowly applied force along the suture line using a walnut clip, divided the inner skin into two halves, and observed the color of the walnut testa under natural light. According to the color changes in the walnut testa of ‘Jinghong 1’, the developmental stages of walnuts were divided into green fruit (G), testa color transition (T), and ripening stages (R). Collected the testa and kernel of ‘D2-1’ and ‘Jinghong 1’ walnuts during the green fruit, testa color transition, and ripening stages, respectively. And collected the young and mature leaves of ‘D2-1’ and ‘Jinghong 1’ walnuts, with three biological replicates set for each material. The plant materials used for tissue specificity experiments were leaves, kernels, stems, and testa of ‘D2-1’ collected on 12 August 2023; and the leaves, kernels, and stems of ‘Jinghong 1’ collected on 12 August 2023, as well as the testa of ‘Jinghong 1’ collected on 25 June and 12 August 2023. The above materials were treated with liquid nitrogen and stored in the −80 °C ultra-low temperature freezer for future use. The walnut seedlings and *Nicotiana benthamiana* used for instantaneous transformation were both 3 weeks old. The growth conditions were set to the photoperiod of 16 h/8 h, and the temperature of 24–25 °C.

### 4.2. Determination of Anthocyanin Content

The anthocyanin content in the young leaves and testa of ‘D2-1’ and ‘Jinghong 1’ was determined using the pH differential method with cyanidin-3-glucoside as the standard. Extracted anthocyanins from the young leaves and testa using the 50% methanol solution containing 0.1% HCl, centrifuged, and removed the supernatant. Two portions of the supernatant, one portion were added to a pH 1.0 potassium chloride buffer solution, and the other portion was added to a pH 4.5 sodium acetate buffer solution. Mix well and let it stand for equilibrium. Measured the absorbance of the two solutions using the UV spectrophotometer (UV-2600, Shimadzu, Kyoto, Japan) at wavelengths of 510 nm and 700 nm, respectively. The formula for calculating the total anthocyanin content is $\;\frac{\left(A510-A700\right)pH1.0-\left(A510-A700\right)pH4.5}{\varepsilon \times L\times m}\times DF\times MW\times V\times 10^{3}$. MW is the molecular weight of 449.2 g/mol. DF is the dilution factor. V is the anthocyanin extract volume. ε is the molar absorptivity of cyanidin-3-glucoside at 26,900 L/mol·cm. L is the optical path of 1cm, and m is the mass of plant material used for anthocyanin determination. Finally, expressed the anthocyanin content in mg/kg (FW).

### 4.3. RNA Extraction, cDNA Library Preparation, and Illumina Sequencing

Following the method of Zhang et al. [40], total RNA was extracted using the FastPure Cell/Tissue Total RNA Isolation Kit V2 kit (Vazyme, Nanjing, China). Removed ribosomal RNA from total RNA to obtain mRNA. Added the fragmentation buffer to break the RNA into short fragments, used the short fragment RNA as the template, and synthesized the first-strand cDNA and the second-strand cDNA separately. And used purified AMPure XP beads double stranded cDNA. The double-stranded cDNA was subjected to terminal repair, followed by the addition of the A tail and the connection of sequencing adapters. The fragment size was selected using AMPure XP beads, and finally, PCR enrichment was performed to obtain the final cDNA library. Preliminary quantification was performed using Qubit (v2.0), and the insert size of the library was detected using Agilent 2100. After the insert size met expectations, transcriptome sequencing was performed using the Illumina sequencing platform (Maiwei Biotechnology Co., Ltd., Wuhan, China).

### 4.4. Quality Control, Functional Annotation, Differential Gene Screening, and Functional Enrichment

Used fastp to remove reads containing connectors and low-quality reads. Used HISAT 2 (v2.2.1) to perform sequence alignment between high-quality clean reads that have undergone quality control and the walnut reference genome. The reference genome is Walnut 2.0 (https://www.ncbi.nlm.nih.gov/datasets/genome/GCF_001411555.2/) (accessed on 27 November 2025). The accession number in the NCBI BioProject database is PRJNA291087 [41]. For new genes, all genes were compared with GO, KEGG, NR, Swiss Prot, trEMBL, and KOG databases using Diamond, and annotation results were obtained under the condition of E-value < 0.05. Based on the comparison results and the location information of genes on the reference genome, quantitative analysis of gene expression levels was performed using FPKM as the quantitative indicator. Using the ‘D2-1’ testa as the control group, DESeq2 (v3.2) was used to analyze the differential gene expression patterns of the ‘Jinghong 1’ testa. The screening criteria for differentially expressed genes between different samples were |Log_2_FC| ≥ 1 and FDR < 0.05. The significant enrichment analysis of pathways was based on KEGG enrichment analysis, using hypergeometric tests to identify pathways with significant enrichment in differentially expressed genes. Displayed the KEGG enrichment results using the scatter plot. Time series and WGCNA analysis were performed using the Baimaike Cloud Platform (https://www.biocloud.net/) (accessed on 18 March 2025).

### 4.5. Metabolite Extraction, Determination, Qualitative and Quantitative Analysis

Accurate quantitative analysis of the testa and kernel of ‘D2-1’ and ‘Jinghong 1’ during the green fruit, testa color transition, and ripening stages was performed using liquid chromatography tandem mass spectrometry (LC-MS/MS) (Maiwei Metabolism, Wuhan, China). Extracted anthocyanin compounds from the test material using the 50% methanol solution containing 0.1% hydrochloric acid. Data collection was performed using Ultra Performance Liquid Chromatography (UPLC, ExionLC™ AD, https://sciex.com.cn/) (accessed on 16 December 2024) and tandem mass spectrometry (MS/MS, QTRAP^®^ 6500+, https://sciex.com.cn/) (accessed on 16 December 2024). Constructed the MWDB database based on standard samples and performed qualitative analysis on mass spectrometry detection data. Quantitative analysis was performed using the Multiple Reaction Monitoring (MRM) mode of triple quadrupole mass spectrometry. Used Analyst (v1.6.3) to process mass spectrometry data. Used MultiQuant (v3.0.3) to process and perform integral correction on mass spectrometry data. Prepared standard solutions with different concentrations of 0.01 ng/mL, 0.05 ng/mL, 0.1 ng/mL, 0.5 ng/mL, 1 ng/mL, 5 ng/mL, 10 ng/mL, 50 ng/mL, 100 ng/mL, 500 ng/mL, 1000 ng/mL, 2000 ng/mL, and 5000 ng/mL. And obtained chromatographic peak intensity data for the corresponding quantitative signals of each concentration standard. Drew standard curves for different substances with concentration as the horizontal axis and peak area as the vertical axis. Substituted the integrated peak area of all detected samples into the linear equation of the standard curve for calculation, and further substituted it into the calculation formula to obtain the actual content data of the substance in the sample.

### 4.6. Differential Metabolite Screening and Transcriptome Metabolome Combined Analysis

Based on the OPLS-DA model (biological replicates ≥ 3), the Variable Importance in Projection (VIP) was obtained to preliminarily screen for differential metabolites between the testa of ‘D2-1’ and ‘Jinghong 1’. The screening criteria for differential metabolites were fold change ≥ 2 and fold change ≤ 0.5, and the difference between the control group and the experimental group was ≥2 or ≤0.5.

Performed correlation analysis using quantitative values of genes and metabolites in all samples. The correlation method used the cor function in R (v4.4.2) to calculate the Pearson correlation coefficient between genes and metabolites. The correlation coefficient ≥ 0.80 and *p*-value ≤ 0.05 was considered to have a significant correlation. Displayed the multiple differences in substances with a Pearson correlation coefficient ≥ 0.80 and *p*-value ≤ 0.05 in each differential group through the nine-quadrant plot. Divided into 1–9 quadrants from left to right and top to bottom using black dashed lines. The 5th quadrant represented non-differential expression of genes and metabolites. The 3rd and 7th quadrants represented consistent differential expression patterns between genes and metabolites. The 1st and 9th quadrants represented opposite differential expression patterns between genes and metabolites. The 2nd, 4th, 6th, and 8th quadrants represented unchanged metabolite expression, gene upregulation or downregulation, or unchanged gene expression, and metabolite upregulation or downregulation.

### 4.7. JrMYB1L Protein Multi-Sequence Alignment and Phylogenetic Tree Analysis

The NCBI CD search online website (https://www.ncbi.nlm.nih.gov/Structure/cdd/wrpsb.cgi) (accessed on 8 April 2025) was used to predict the structural domains of JrMYB1L. Jalview (v2.11.2.3) and MEGA11 (v11.0.13) were used for multiple sequence alignment of JrMYB1L protein. Downloaded R2R3-MYB protein sequences of walnut JrMYB1L, *Arabidopsis*, *Chaenomeles speciosa*, peach, and other species from the NCBI database (https://www.ncbi.nlm.nih.gov) (accessed on 8 April 2025). Based on the UPGMA method, the phylogenetic tree was constructed using MEGA11 (v11.0.13) to investigate the relationship between the JrMYB1L protein and R2R3-MYB proteins in *Arabidopsis*, American walnut, European hazelnut, *Chaenomeles speciosa*, peach, and other plants.

### 4.8. qRT-PCR

RNA was reverse transcribed into cDNA using a reverse transcription kit (Vazyme, Nanjing, China), and its concentration was measured using an ultramicro visible ultraviolet spectrophotometer (ND-5000, National Diagnostics, Georgia, American). Primers for *JrMYB1L*, *JrCHS*, *JrCHI*, *JrF3H*, *JrDFR*, *JrANS*, and *JrUFGT* were designed using Primer 3 Plus (https://www.primer3plus.com) (accessed on 23 June 2024). Real-time fluorescence quantitative PCR (qRT-PCR) was performed using the Borui LineGene9600plus fluorescence quantitative PCR instrument. Using the ChamQ SYBR Color qPCR Master Mix (2×) kit (Vazyme, Nanjing, China), the reaction process was as follows: 95 °C for 3 min, one cycle; 95 °C for 30 s, 56 °C for 30 s, 72 °C for 40 s, 35 cycles; followed by reading Ct values and calculating relative expression levels. *18S* was the reference gene for this experiment, and the relative expression level of the target gene was calculated using the 2^−ΔΔCt^ method. The relative expression level of the target gene was 2^−ΔΔCt^, and the experiment was repeated 3 times. Appendix A lists the qRT-PCR primers for anthocyanin biosynthesis pathway genes.

### 4.9. Transient Overexpression of JrMYB1L

Following the method of Zhang et al. [40], the full-length CDS sequence of *JrMYB1L* was cloned, and the primers are listed in Appendix A. Double enzyme cleavage of the pCAMBIASuper1300 vector was performed using XbaI and KpnI enzymes (Takara, Beijing, China). The CDS sequence of *JrMYB1L* was inserted into the pCAMBIASuper1300 linearized vector using homologous recombinase (Vazyme, Nanjing, China). Constructed pCAMBIASuper1300-*JrMYB1L* plasmid and transferred it into GV3101 competent cells. After centrifuging the overnight cultured bacterial solution at 5000 rpm for 10 min, the bacterial cells were resuspended in infection solution (10 mM MgCl_2_ + 10 mM MES + 50 mg/L AS, pH = 5.6) until the OD was 0.6~0.8. Let the pCAMBIASuper1300-*JrMYB1L* infection solution stand in the dark at room temperature for 3–4 h. Healthy walnut rooting seedlings were used as test materials, soaked in infection solution, and the infection solution without Agrobacterium was used as a blank control. Three biological replicates were set for each treatment. Placed it under negative pressure of −0.10–0.09 MPa for 30 min, then slowly restored it to normal pressure. After removing the walnut seedlings, rinsing them three times with tap water, and planting them in the mixture of peat soil–vermiculite = 1:1. Let them stand in the dark at room temperature for 24 h, then incubate them under light for 5 days. Took samples after 5 days.

### 4.10. Self-Activation Verification and Yeast Two-Hybrid

Y2HGold competent cells were constructed by Beijing Wanjing Inspirational Biotechnology Co., Ltd. (Beijing, China). Single enzyme digestion was performed on pGBKT7 (BD) and pGADT7 (AD) vectors using BamHI enzyme (Takara, Beijing, China). The CDS sequences of JrbHLH42 and JrWD40 were inserted into the pGBKT7 vector using homologous recombinase (Vazyme, Nanjing, China). Co-transferred with pGADT7 empty vector to Y2HGold competent cells for self-activation validation. The primers were listed in Appendix A. Inserted the CDS sequences of JrMYB1L and JrbHLH42 into the pGADT7 vector separately. Transferred JrbHLH42-BD + JrMYB1L-AD, JrWD40-BD + JrMYB1L-AD, and JrWD40-BD + JrbHLH42-AD to Y2HGold competent cells separately to verify the interaction between two proteins. According to the Y2HGold Chemically Competitive Cell manual from Shanghai Weidi Biotechnology Co., Ltd. (Shanghai, China), the bait protein and prey protein were co-transformed into Y2HGold competent cells. The co-transformed bacterial solution was coated onto the two deficient medium (SD/-Trp/-Leu) and cultured at 28 °C for 2–3 days. Single colonies were then selected for expansion culture. Subsequently, the expanded bacterial solution was diluted in gradients of 10^0^, 10^−1^, 10^−2^, and 10^−3^, respectively. Diluted the bacterial solution separately and placed it on two deficiency medium (SD/-Trp/-Leu), three deficiency medium (SD/-His/-Trp/-Leu), and four deficiency medium (SD/-His/-Ade/-Trp/-Leu/). After culturing at 28 °C for 3–5 days, we observed and took photos.

### 4.11. Yeast One-Hybrid

Y1HGold sensory cells were constructed by Beijing Wanjing Inspirational Biotechnology Co., Ltd. (Beijing, China). Single enzyme digestion of pGADT7 and pAbAi vectors was performed using BamHI enzyme (Takara, Beijing, China). Connected the full-length *JrMYB1L* with the pGADT7 linearized vector to construct the pGADT7-*JrMYB1L* plasmid. Connected *JrCHSpro1427-1509* and *JrUFGTpro1200-1400* fragments to the pAbAi linearized vector to construct pAbAi-*JrCHSpro1427-1509* and pAbAi-*JrUFGTpro1200-1400* plasmids. Linearization of the above plasmids was performed using the BstBI enzyme, and then transferred into Y1HGold competent cells to generate Y1HGold competent cells carrying *JrCHSpro 1427-1509* and *JrUFGTpro 1200-1400* fragments. Transferred pGADT7 empty vector into Y1HGold competent cells carrying *JrCHSpro 1427-1509* and *JrUFGTpro 1200-1400*, and coated them on SD/-Ura medium to determine the self-activation activity of *JrCHSpro 1427-1509* and *JrUFGTpro 1200-1400*. Added 0 ng/mL, 100 ng/mL, 200 ng/mL, 300 ng/mL, 500 ng/mL, 700 ng/mL, 850 ng/mL, and 1000 ng/mL of AbA to SD/-Ura medium (Beijing Wanjing Inspirational Biotechnology Co., Ltd., Beijing, China). Selected concentrations that can inhibit the self-activation of *JrCHSpro 1427-1509* and *JrUFGTpro 1200-1400.* After determining the concentration, the pGADT7-*JrMYB1L* plasmid was introduced into Y1HGold competent cells carrying *JrCHSpro1427-1509* and *JrUFGTpro1200-1400.* And Coated on SD/-Leu medium and SD/-Leu medium with the corresponding concentration of AbA added. After culturing at 28 °C for 3–5 days, we observed the binding situation and took photos.

### 4.12. Dual Luciferase

The method for obtaining pGreenII 0800-LUC (LUC)-*JrCHSpro1427-1509* and LUC-*JrUFGTpro1200-1400* plasmids was the same as Section 4.11. Transferred pGreenII 62-SK (SK) empty load, SK-*JrMYB1L*, LUC-*JrCHSpro1427-1509*, and LUC-*JrUFGTpro1200-1400* into GV3101 competent cells. The method for obtaining the infection solution was the same as 4.9. Mixed LUC-*JrCHSpro1427-1509* and LUC-*JrUFGTpro1200-1400* infection solutions with SK and SK-*JrMYB1L* infection solutions in the 1:1 ratio. Injected SK-empty + LUC-*JrCHSpro1427-1509* and SK-*JrMYB1L* + LUC-*JrCHSpro1427-1509* separately on the left and right sides of the three leaves of Nicotiana benthamiana. Injected SK-empty + LUC-*JrUFGTpro1200-1400* and SK-*JrMYB1L* + LUC-*JrUFGTpro1200-1400* on the left and right sides of the three leaves of another plant. Set three biological replicates and three technical replicates for each combination. Stored at room temperature in the dark for 24 h, then incubated under light for 2 days. Applied 1 × D-fluorescein potassium salt (Beijing Wanjing Inspirational Biotechnology Co., Ltd., Beijing, China) evenly at the injection site. Observed fluorescence signals using the protein luminescence imaging system (Beijing Yuanpinghao Biotechnology Co., Ltd., Beijing, China). Used the Dual Luciferase Reporter Assay Kit (Vazyme, Nanjing, China) and the multifunctional enzyme-linked immunosorbent assay (Promega, Beijing, China) to detect the activity of the reported genes for the combination of Firefly Luciferase and Renilla Luciferase.

### 4.13. Data Analysis

SPSS 26.0 software was used for statistical analysis of all data. The *t*-test was used for the analysis of variance between two sets of data. The data were the average of three biological replicates, with error bars representing standard error. Origin 22.0 was used for drawing relevant result graphs. TBtools-Ⅱ (v2.097) were used for drawing heatmaps.

## 5. Conclusions

The anthocyanin content in the testa of ‘Jinghong 1’ in all stages was significantly higher than that of ‘D2-1’, and the main components were cyanidin-3-O-arabinoside, cyanidin-3-O-galactoside, and cyanidin-3-O-glucoside. KEGG enrichment, time-series, and WGCNA results indicated that differentially expressed genes between ‘D2-1’ and ‘Jinghong 1’ were mainly enriched in the phenylpropanoid biosynthesis and flavonoid biosynthesis pathways. A TT2-type *R2R3-MYB* transcription factor *JrMYB1L* (*LOC109001985*) was screened, and the regulatory mechanism of anthocyanin biosynthesis in the testa of ‘Jinghong 1’ was proposed. JrMYB1L, JrbHLH42, and JrWD40 proteins together formed a classic MBW complex. The *JrMYB1L* transcription factor activated its expression by binding to the *JrCHS* and *JrUFGT* promoters, thereby promoting anthocyanin accumulation. These results not only lay the foundation for clarifying the regulatory mechanism between TT2-type *R2R3-MYB* transcription factors and anthocyanin biosynthesis but also provide a theoretical basis for breeding high-quality red walnuts.

## Figures and Tables

**Figure 1 plants-14-03727-f001:**
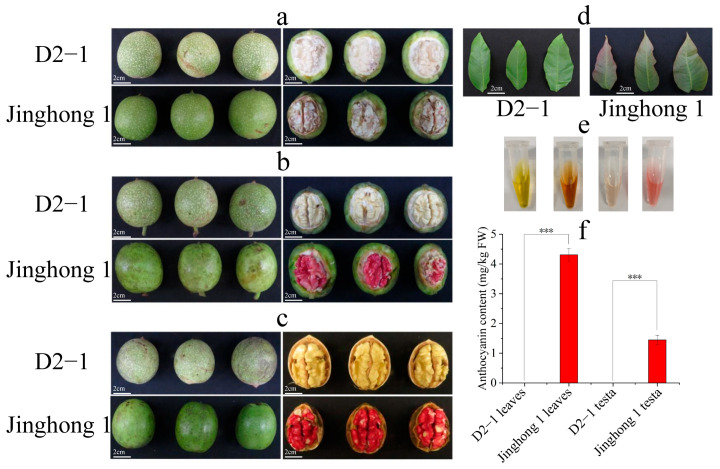
Phenotypic traits and anthocyanin content of ‘D2-1’ and ‘Jinghong 1’ walnut young leaves and testa. (**a**) Phenotypic traits of walnut fruits in the green fruit stage. (**b**) Phenotypic traits of walnut fruits in the testa color transition stage. (**c**) Phenotypic traits of walnut fruits in the ripening stage. (**d**) Phenotypic traits of young leaves in ‘D2-1’ and ‘Jinghong 1’ walnuts. (**e**) Anthocyanin extract from young leaves and testa (in the ripening stage) of ‘D2-1’ and ‘Jinghong 1’. From left to right, they represent young leaves of ‘D2-1’, young leaves of ‘Jinghong 1’, the testa of ‘D2-1’, and the testa of ‘Jinghong 1’. (**f**) Anthocyanin content in young leaves and testa (in the ripening stage) of ‘D2-1’ and ‘Jinghong 1’. *** represents a highly significant difference in anthocyanin content between samples.

**Figure 2 plants-14-03727-f002:**
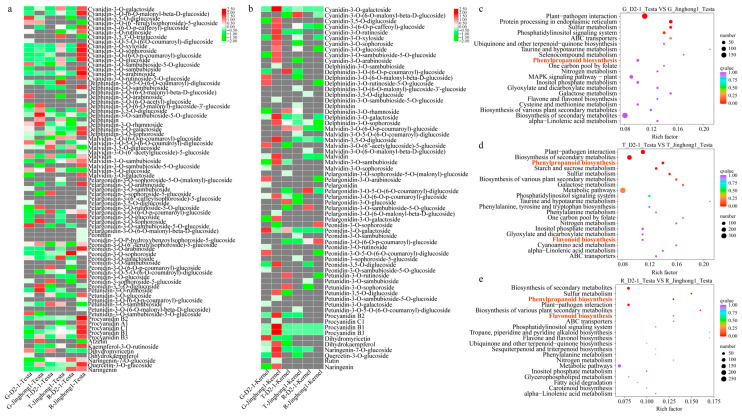
Differential genes and metabolites in the testa of ‘D2-1’ and ‘Jinghong 1’. (**a**) The differential metabolites between the testa of ‘D2-1’ and ‘Jinghong 1’. (**b**) The differential metabolites between the kernel of ‘D2-1’ and ‘Jinghong 1’. G, T, and R represent the green fruit, testa color transition, and ripening stages, respectively. (**c**–**e**) The KEGG enrichment results of differentially expressed genes in the testa of ‘D2-1’ and ‘Jinghong 1’ in the green fruit (G), testa color transition (T), and ripening (R) stages, respectively.

**Figure 3 plants-14-03727-f003:**
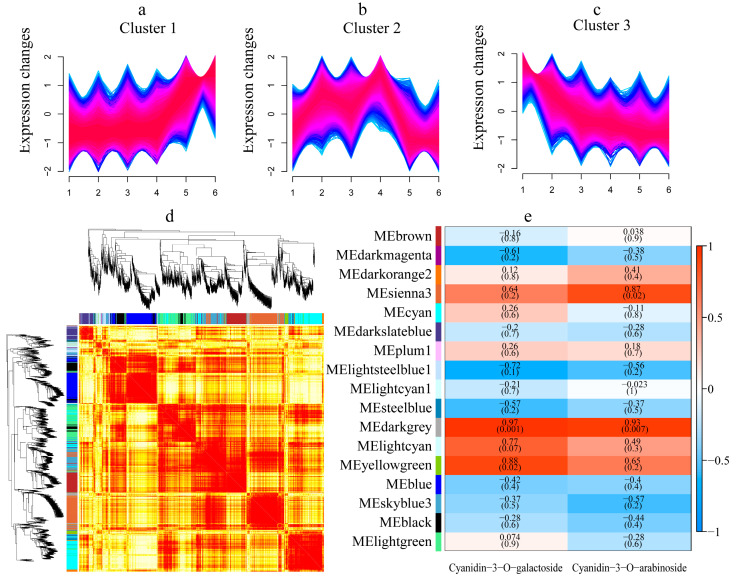
Time series analysis and WGCNA results. (**a**–**c**) represented cluster 1~cluster 3, respectively. 1~2, 3~4, and 5~6 represented ‘D2-1’ testa and ‘Jinghong 1’testa in the green fruit, testa color transition, and ripening stages, respectively. (**d**) Gene cluster dendrogram and the correlation between modules. (**e**) The correlation between modules and anthocyanin content. The number above represented the coefficient, and the number below represented R^2^ value. Red represents positive correlation, while blue represents negative correlation.

**Figure 4 plants-14-03727-f004:**
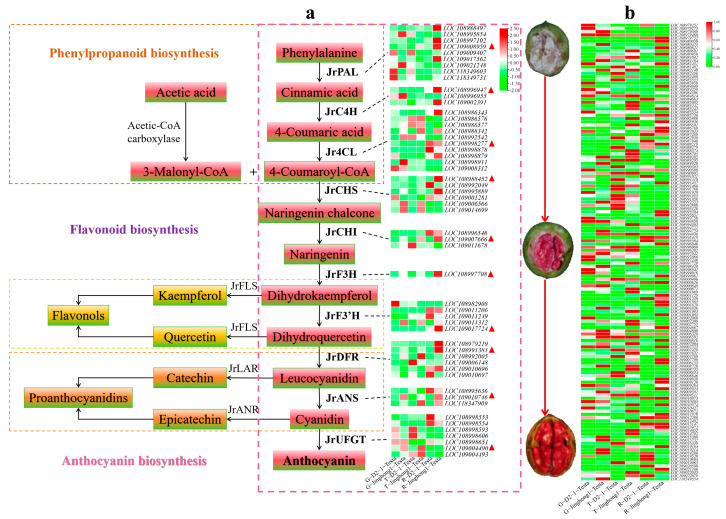
Expression characteristics of anthocyanin biosynthesis pathway structural genes (**a**) and *JrMYB* transcription factor (**b**). G, T, and R represent green fruit, testa color transition, and ripening stages, respectively. Red triangles in the figure represent the selected key differentially expressed genes.

**Figure 5 plants-14-03727-f005:**
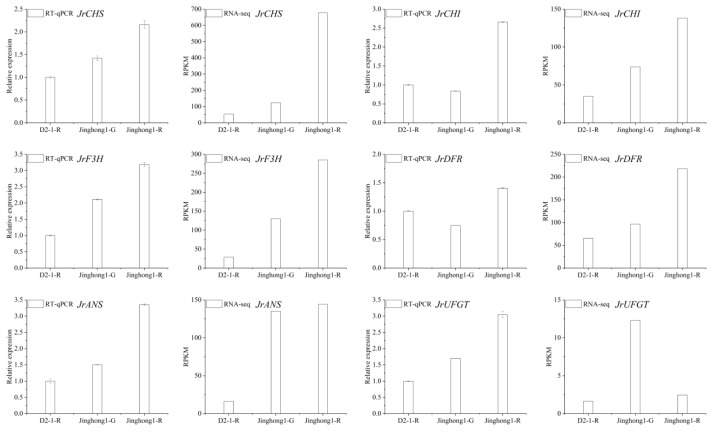
Expression characteristics of key genes in the anthocyanin biosynthesis pathway in the testa of ‘D2-1’ and ‘Jinghong 1’. The **left** figure shows the RT-qPCR results, and the **right** figure shows the RNA-seq results. D2-1-R, Jinghong 1-G, and Jinghong 1-R represent the testa of ‘D2-1’ in the ripening stage, the testa of ‘Jinghong 1’ in the green fruit and ripening stages, respectively.

**Figure 6 plants-14-03727-f006:**
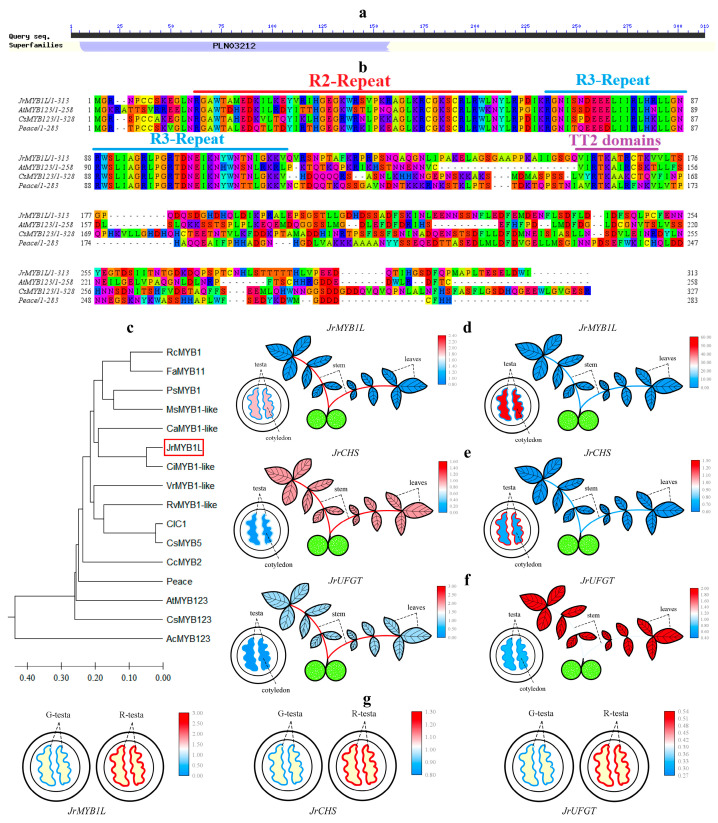
Bioinformatics analysis of *JrMYB1L* and tissue specificity of *JrMYB1L, JrCHS,* and *JrUFGT.* (**a**) Structural domain prediction results of the JrMYB1L protein. (**b**) Multiple sequence alignment results between JrMYB1L, AtMYB123, CsMYB123, and Peace proteins. The same color represents the homology of amino acid sequences. (**c**) Phylogenetic relationship between JrMYB1L and TT2 type R2R3-MYB proteins in other species. The red box represents the key gene *JrMYB1L* in this article. (**d**–**f**) The expression characteristics of *JrMYB1L, JrCHS,* and *JrUFGT* in different tissues of ‘D2-1’ (**left**) and ‘Jinghong 1’ (**right**). (**g**) The expression characteristics of *JrMYB1L* (**left**), *JrCHS* (**middle**), and *JrUFGT* (**right**) in the testa of ‘Jinghong 1’ in the green fruit (G) and ripening (R) stages. In (**d**–**f**), red represents relatively high expression levels, while red represents relatively low expression levels.

**Figure 7 plants-14-03727-f007:**
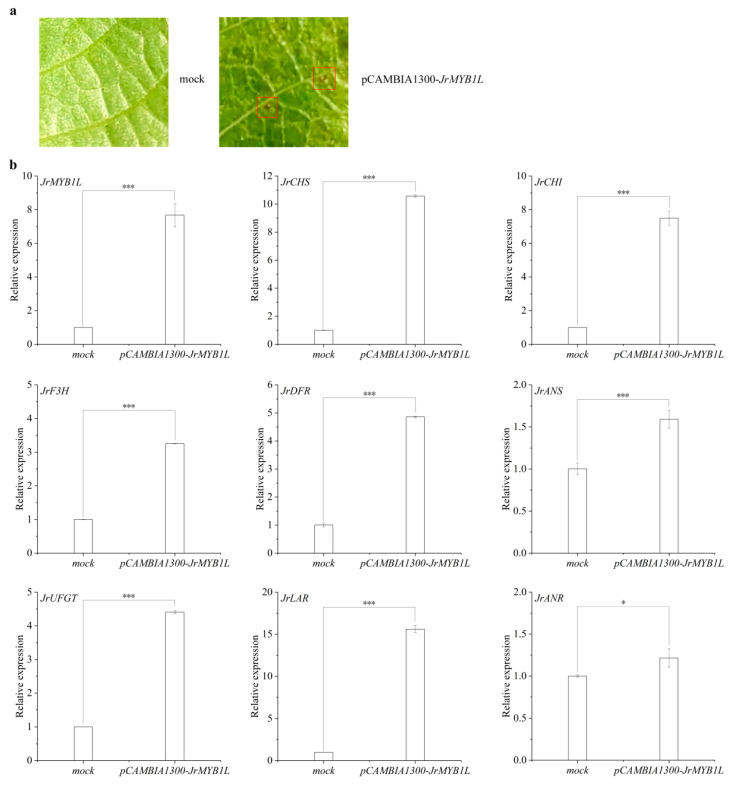
*JrMYB1L* transient overexpression participates in regulating anthocyanin biosynthesis. (**a**) Phenotypic traits of walnut leaves transiently overexpressed with *JrMYB1L*. The red box represents the accumulation of anthocyanins. (**b**) Expression changes in anthocyanin and proanthocyanidin biosynthesis structural genes in *JrMYB1L* transiently overexpressing walnut leaves. The values in b represent the means ± standard deviations of three biological replicates. Based on the results of the two-sample *t*-test analysis, * represents *p* < 0.05, and *** represents *p* < 0.01.

**Figure 8 plants-14-03727-f008:**
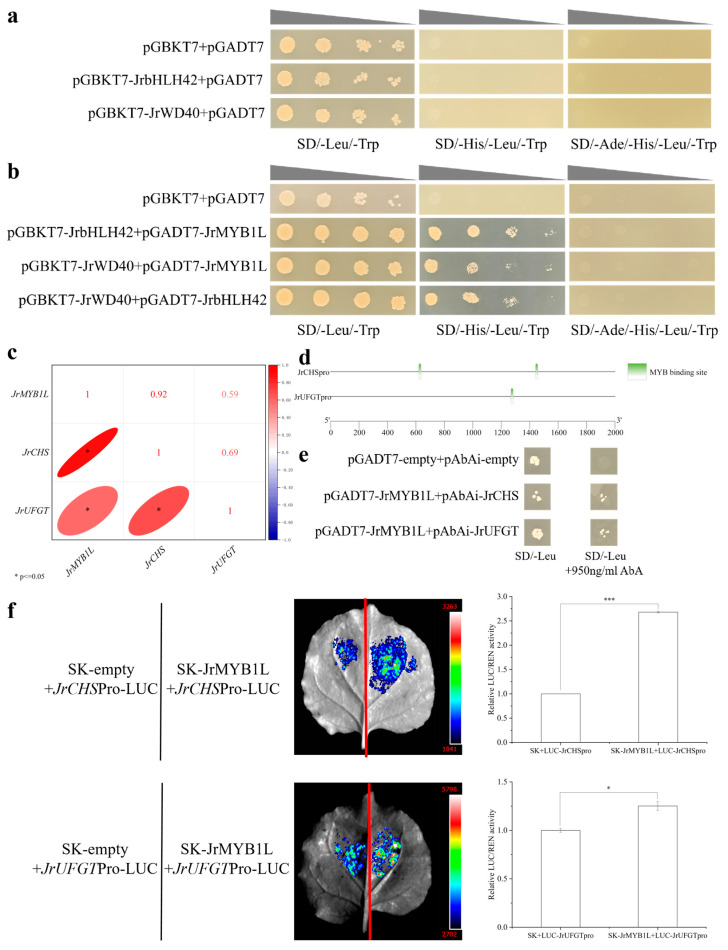
The binding of JrMYB1L interacting proteins and their downstream target genes. (**a**) Self-activation of JrbHLH42 and JrWD40 proteins. (**b**) Y2H confirms the interaction between JrMYB1L, JrbHLH42, and JrWD40 proteins. (**c**) The correlation between transcription levels of *JrMYB1L*, *JrCHS*, and *JrUFGT*. (**d**) Distribution of binding sites between *JrCHSpro*, *JrUFGTpro* fragments, and *JrMYB1L*. (**e**) Y1H proves the binding of *JrMYB1L* with *JrCHSpro* and *JrUFGTpro*. (**f**) Dual-LUC demonstrates the binding of *JrMYB1L* with *JrCHSpro* and *JrUFGTpro*, with qualitative and quantitative analysis results from left to right. The values in f represent the means ± standard deviations of three biological replicates. Based on the results of the two-sample *t*-test analysis, * represents *p* < 0.05, and *** represents *p* < 0.01.

**Figure 9 plants-14-03727-f009:**
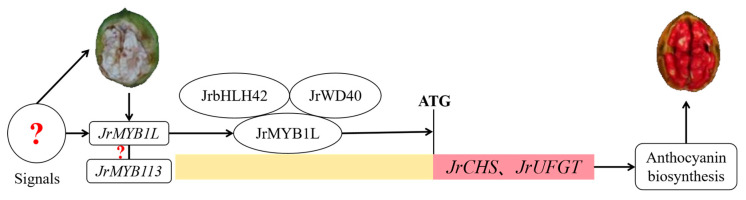
Regulation mechanism of anthocyanin biosynthesis in the testa of ‘Jinghong 1’. Red question marks represent the unknown relationship.

## Data Availability

All data supporting the findings of this study are available in the manuscript and online Appendix A.

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
