# Peer review of "The Mechanism of TT2-Type MYB Transcription Factor JrMYB1L in Anthocyanin Biosynthesis in ‘Jinghong 1’ Walnuts"

_plants, 2025, doi:10.3390/plants14243727_

Round 1
Reviewer 1 Report
Comments and Suggestions for Authors
This manuscript addresses the mechanistic role of JrMYB1L in controlling walnut testa coloration. The study represents a considerable body of work and its findings are of notable interest. However, the clarity and overall presentation of the manuscript require significant enhancements to match the scientific quality of the research. The abstract requires significant revision to more effectively frame the study's background and to articulate the research conclusions with greater clarity. Specifically, the background fails to clearly outline the research gap, while the conclusions lack precision. Furthermore, the use of identical phrasing (e.g., in lines 22-24 and 25) to describe distinct experimental outcomes is confusing and undermines the clarity of the findings.. The introduction has a key problem: it reads like a literature review. It must clearly state the research background, the unresolved question, and the purpose of this work. The first paragraph is too long—cut it down to just introduce the materials and their traits. Revise the rest to be more focused on your own study. The conclusion drawn in lines 176-177, based solely on the results from lines 169-170, constitutes an overinterpretation. To robustly identify the primary pigments responsible for coloration, the analysis should be strengthened by either: 1) quantifying the percentage composition of individual anthocyanin components, or 2) performing a correlation analysis with the colorimetric values (L, a, b*). In Figure 2a, metabolites that were undetected across all samples should be removed. Figures 2b–d lack practical significance, as they merely indicate the number of differentially expressed genes without providing clear insights. The overall clarity of these figures needs improvement. Several editorial errors should be addressed, such as "8" in line 210, "3.1" in line 263, and "3.3" in line 269. The analysis of gene expression across different plant tissues (lines 286-299) would be much stronger if correlated with anthocyanin levels from those same tissues. In the absence of such correlative data, this section lacks context and relevance. To maintain focus, the presentation of expression data should be confined to the testa, which is the primary subject of this study. In Section 2.6, Figure 6a does not convincingly demonstrate anthocyanin accumulation. It is recommended to supplement this with microscopic images or additional quantitative data on anthocyanin content. The evidence presented in Figure 6a for anthocyanin accumulation in Section 2.6 is currently not compelling. To substantiate this key claim, it is essential to provide more direct evidence, such as microscopic images showing pigment localization or quantitative data from spectrophotometric assays. In lines 443–457, there is inconsistency regarding whether transient expression of JrMYB1L leads to anthocyanin accumulation in leaves. This discrepancy should be clarified. The sentence in lines 460–461 is poorly constructed and does not meet academic writing standards.
Author Response
Reviewer 1
This manuscript addresses the mechanistic role of JrMYB1L in controlling walnut testa coloration. The study represents a considerable body of work and its findings are of notable interest. However, the clarity and overall presentation of the manuscript require significant enhancements to match the scientific quality of the research.
Thank you for your recognition of our research. We have fully considered your suggestions and made revisions to the manuscript. All modifications are highlighted. The revision has been resubmitted to the journal. We look forward to your positive response.
- The abstract requires significant revision to more effectively frame the study's background and to articulate the research conclusions with greater clarity. Specifically, the background fails to clearly outline the research gap, while the conclusions lack precision. Furthermore, the use of identical phrasing (e.g., in lines 22-24 and 25) to describe distinct experimental outcomes is confusing and undermines the clarity of the findings.
Response: Thank you for your suggestion. We have added a description of the background and significance of this study, as shown in lines 15-16. In addition, we have revised the wording of the results, as shown in lines 26-29.
- The introduction has a key problem: it reads like a literature review. It must clearly state the research background, the unresolved question, and the purpose of this work. The first paragraph is too long—cut it down to just introduce the materials and their traits. Revise the rest to be more focused on your own study.
Response: Thank you for your suggestion. We have removed irrelevant descriptions in the first paragraph of the introduction, as is shown in lines 37-47. In addition, some redundant reviews have been removed.
- The conclusion drawn in lines 176-177, based solely on the results from lines 169-170, constitutes an overinterpretation. To robustly identify the primary pigments responsible for coloration, the analysis should be strengthened by either: 1) quantifying the percentage composition of individual anthocyanin components, or 2) performing a correlation analysis with the colorimetric values (L, a, b*).
Response: Thank you for your suggestion. We have added the proportion data of various types of anthocyanin metabolites in Table S1. In addition, we have added a description of the proportion, as shown in lines 165-179.
- In Figure 2a, metabolites that were undetected across all samples should be removed.
Response: Thank you for your suggestion. We have removed the metabolites that were not detected in Figure 2a.
- Figures 2b–d lack practical significance, as they merely indicate the number of differentially expressed genes without providing clear insights. The overall clarity of these figures needs improvement.
Response: Thank you for your suggestion. We removed Figures 2b-d from the main text and added Figure S1. In addition, we have redrawn Figure 2 to increase the resolution.
- Several editorial errors should be addressed, such as "8" in line 210, "3.1" in line 263, and "3.3" in line 269.
Response: Thank you for your suggestion. The number 8 represents the literature, and we have completed the modification as shown in line 241. We have revised ‘3.1’ and ‘3.3’ to ‘2.1’ in line 299 and ‘2.3’ in line 304.
- The analysis of gene expression across different plant tissues (lines 286-299) would be much stronger if correlated with anthocyanin levels from those same tissues. In the absence of such correlative data, this section lacks context and relevance. To maintain focus, the presentation of expression data should be confined to the testa, which is the primary subject of this study.
Response: Thank you for your suggestion. We conducted correlation analysis using anthocyanin and expression data from seed coat and kernel, and added the correlation between anthocyanin content and JrMYB1L, JrCHS, and JrUFGT in Figure S3. And corresponding descriptions were added in lines 335-337 of the manuscript.
- In Section 2.6, Figure 6a does not convincingly demonstrate anthocyanin accumulation. It is recommended to supplement this with microscopic images or additional quantitative data on anthocyanin content. The evidence presented in Figure 6a for anthocyanin accumulation in Section 2.6 is currently not compelling. To substantiate this key claim, it is essential to provide more direct evidence, such as microscopic images showing pigment localization or quantitative data from spectrophotometric assays.
Response: Thank you for your suggestion. As the sample has already been used for expression level determination, we are unable to take another micrograph. We have added the original photo in Figure S4. Unfortunately, the accumulation of red pigment is not particularly noticeable. We hope to identify more critical genes in future research that can play a more prominent role in anthocyanin biosynthesis.
- In lines 443–457, there is inconsistency regarding whether transient expression of JrMYB1L leads to anthocyanin accumulation in leaves. This discrepancy should be clarified.
Response: Thank you for your suggestion. We have reviewed this section of the description and made some adjustments. The transient overexpression of JrMYB1L resulted in a small amount of red dots appearing in walnut leaves, but no accumulation of anthocyanins was observed in large patches. We provided a detailed explanation on lines 488-501.
- The sentence in lines 460–461 is poorly constructed and does not meet academic writing standards.
Response: Thank you for your suggestion. We have rephrased this sentence, as shown in lines 507-509.
Reviewer 2 Report
Comments and Suggestions for Authors
You will find the comments in the attached file. Thank you and best regards.

Author Response
In this manuscript, the authors explore the mechanisms underlying the regulation and synthesis of anthocyanins in red walnuts. Specifically, they identify the JrMYB1L gene, a TT2-type R2R3-MYB gene that, by interacting with JrbHLH42 and JrWD40, leads to the activation of several anthocyanin biosynthesis genes, primarily cyanidin. The article is well-written, well-structured, and experimentally complete. The results are well-presented, although they could be improved and adequately discussed. The manuscript is not particularly innovative, but considering the species the authors chose, it certainly lays the foundation for further exploration of the field. Below are some considerations to improve understanding of the manuscript.
Response: Thank you for recognizing our research. We also hope that this study can provide a foundation for further research in the future. Thank you very much for your constructive feedback on this manuscript. We have completed all the modifications, as shown below.
- Most figures are composed of numerous parts. These are often not labeled with letters, for example, to adequately distinguish the various parts, often creating confusion. An example is Figure 1. Panel a consists of 14 photos, distinct for the two samples D2.1 and Jinhong 1, the husk and the cross-section, of three different stages indicated across the two sets of photos only by G, T, and R. Panel a also includes two other photos of leaves, thus also a different tissue. When reading the results, as well as the photo captions, it is not always easy to find what you are reading. This applies to most of the figures.
Response: Thank you for your suggestion. We have rearranged the image in Figure 1 and added subheadings. In addition, a detailed description of the image was provided, as is shown in lines 144-153.
- Image quality issues. In the provided file, most of the figures are of poor quality, often illegible, as in the case of Figure 2, where no text is clearly visible. Check whether the quality meets the required level.
Response: Thank you for your suggestion. We have redrawn the image, enlarged the font, and improved the quality of the image, as shown in lines 198-205.
- Figure 5d consists of three parts, but they are not clearly indicated and also leaves the right part in the legend. (lines 305 and 306) Write what is on the left and in the middle, but not what is on the right.
Response: Thank you for your suggestion. We have rearranged this image and provided detailed explanations for each image, as shown in lines 344-347.
- Figure 6a, the two portions of the leaf are of very poor quality, and the colored spots are truly difficult to even guess.
Response: Thank you for your suggestion. We have added the original photo in Figure S4. Unfortunately, the accumulation of red pigment is not particularly noticeable.
- In the text, always write Fig.1a (for example). Please consider the format required by the journal. This applies to the entire body of the manuscript. A space is required after the dot.
Response: Thank you for your suggestion. We have reviewed the entire document and completed the revisions.
- Line 210: …biossynthesis.8 ……What is the number 8? Does it refer to the reference?Then introduce it appropriately. This applies to lines 214 to 219, where it reads (3), The same In Line 244.
Response: Thank you for your suggestion. The number 8 represents the literature, and we have completed the modification as shown in line 241. The numbers on lines 214-219 and 244 represent the number of genes. To avoid ambiguity, we have rephrased this section as shown in lines 244-252.
- Line 363 reports Fig.1, which is not correct.
Response: Thank you for your suggestion. We have completed the revisions in the manuscript, as shown in lines 404-406.
- Line 375 refers to Fig.7g. The relevant panel is not indicated in Figure 7, nor is it even mentioned in the caption.
Response: Thank you for your suggestion. We have completed the revisions in the manuscript, as shown in line 420.
- Discussion section. Please check whether the reference figures can be reported according to the journal or not.
Response: Thank you for your suggestion. We have made further revisions to the literature cited in the discussion section based on the editor's feedback, reducing the self citation rate. Specifically, as shown in lines 809-927.
Reviewer 3 Report
Comments and Suggestions for Authors
This is a well-executed study that utilizes transcriptomic data to identify key genes regulating walnut coloration.
However, it should be noted that the conserved function and mechanism of the MBW complex in regulating anthocyanin accumulation across various organisms may impact the novelty of this research.
Therefore, a more systematic analysis of transcriptomic data is recommended to uncover novel core regulatory factors. The functional investigation of these factors in the coloration process should focus on genes specific to this material.
Additionally, the authors should provide a table of expression levels for all genes (in FPKM or TPM format), along with clear annotations of the referenced genome and the corresponding accession numbers for each gene. The reference genomes to be used are:
· Chandler v1.0 (as referenced in the study The walnut (Juglans regia) genome sequence reveals diversity in genes coding for the biosynthesis of non-structural polyphenols),
· or Chandler v2.0 (as referenced in the study High-quality chromosome-scale assembly of the walnut (Juglans regia L.) reference genome).
Author Response
This is a well-executed study that utilizes transcriptomic data to identify key genes regulating walnut coloration. However, it should be noted that the conserved function and mechanism of the MBW complex in regulating anthocyanin accumulation across various organisms may impact the novelty of this research.
Response: Thank you very much for your recognition of our research. We have made revisions to the manuscript based on your feedback.
- Therefore, a more systematic analysis of transcriptomic data is recommended to uncover novel core regulatory factors. The functional investigation of these factors in the coloration process should focus on genes specific to this material.
Response: Thank you for your suggestion. We have supplemented the systematic analysis of the transcriptome by incorporating the results of time-series analysis and WGCNA analysis, as shown in lines 206-237.
- Additionally, the authors should provide a table of expression levels for all genes (in FPKM or TPM format), along with clear annotations of the referenced genome and the corresponding accession numbers for each gene. The reference genomes to be used are:
- Chandler v1.0 (as referenced in the study The walnut (Juglans regia) genome sequence reveals diversity in genes coding for the biosynthesis of non-structural polyphenols),
- or Chandler v2.0 (as referenced in the study High-quality chromosome-scale assembly of the walnut (Juglans regia L.) reference genome).
Response: Thank you for your suggestion. We have added the expression levels and annotation results of genes mentioned in the manuscript in the appendix, in Supplementary Table S9.
Round 2
Reviewer 1 Report
Comments and Suggestions for Authors
An article should not simply list all the data analyses conducted but rather present conclusions through step-by-step data-driven arguments. It is hoped that the author will exercise discretion in selecting data for future writing, ensuring a logical progression and coherence in the presentation.
Author Response
Comment: An article should not simply list all the data analyses conducted but rather present conclusions through step-by-step data-driven arguments. It is hoped that the author will exercise discretion in selecting data for future writing, ensuring a logical progression and coherence in the presentation.
Response: We appreciate your critical suggestions. In order to improve the readability and logicality of the manuscript, we read the entire manuscript thoroughly. We added descriptions to enhance logic and highlighted them. We hope this revision has resolved the logical sequence issue before and after the manuscript. Furthermore, the limitations of the experimental results in this study have also been mentioned in the discussion section. Although some of the experimental results in this study were not perfect, we still hope to lay a certain foundation for future research. In future writing, we will pay attention to the constructive suggestions you provide. Thank you sincerely again for your review.
Reviewer 3 Report
Comments and Suggestions for Authors
Based on the genome presented by Marrano et al., more than 30,000 genes were annotated (with gene accession numbers such as Gene_ID: Jr01_00010). The authors were required to upload expression levels (read count is particularly required to be presented and FPKM or TPM values) for all genes in the transcriptome, but Supplementary Table S9 only displays limited information for approximately 200 genes. It is strongly recommended that the authors supplement the data to enhance the credibility of the research findings.
Reference genome: Chandler v2.0
Reference: Marrano A, Britton M, Zaini PA, Zimin AV, Workman RE, Puiu D, Bianco L, Pierro EAD, Allen BJ, Chakraborty S, Troggio M, Leslie CA, Timp W, Dandekar A, Salzberg SL, Neale DB. High-quality chromosome-scale assembly of the walnut (Juglans regia L.) reference genome. Gigascience. 2020, (5):giaa050. doi: 10.1093/gigascience/giaa050.
Author Response
Comment: The authors were required to upload expression levels (read count is particularly required to be presented and FPKM or TPM values) for all genes in the transcriptome, but Supplementary Table S9 only displays limited information for approximately 200 genes. It is strongly recommended that the authors supplement the data to enhance the credibility of the research findings.
Response: Thank you for your suggestion. We have updated the content in Supplementary Table S9, providing the FPKM values of all genes in the 'D2-1' and 'Jinghong 1' testa, as well as the annotation results of genes in Chandler v2.0. In addition, we also cited the genomic literature you provided, as shown in lines 617 and 951-953. We hope that this data can provide more assistance for the study of the regulation mechanism of anthocyanin biosynthesis. We would like to express our gratitude once again for your review.
Round 3
Reviewer 3 Report
Comments and Suggestions for Authors
We appreciate the author's response and the revisions made to the manuscript. Meanwhile, I have noted that in the submitted table of gene expression levels, approximately 30,000 genes (with corresponding LOC numbers) were annotated, including over 3,000 novel genes. This raises the question: based on the genome version used by the authors, does this imply that more than 10% of the sequencing results failed to align with the existing reference genome?
The authors claim to have used the Chandler v2.0 genome sequencing results as a reference, and the annotated gene ID format should follow the pattern of Jr01_00010, etc. To ensure data reliability, we strongly recommend that the authors re-submit the information in Supplementary Table S9 for verification.
Author Response
Comment: I have noted that in the submitted table of gene expression levels, approximately 30,000 genes (with corresponding LOC numbers) were annotated, including over 3,000 novel genes. This raises the question: based on the genome version used by the authors, does this imply that more than 10% of the sequencing results failed to align with the existing reference genome?The authors claim to have used the Chandler v2.0 genome sequencing results as a reference, and the annotated gene ID format should follow the pattern of Jr01_00010, etc. To ensure data reliability, we strongly recommend that the authors re-submit the information in Supplementary Table S9 for verification.
Response: Thank you very much for your question. We checked the quality testing results of the transcriptome and found that the error rate of the test samples was 0.03% (Figure 1). The number of reads that can be compared to the reference genome is above 90% (Figure 2). This indicates that our transcriptome results are reliable. Regarding the question you mentioned about the gene ID format being Jr01_00010, we will answer it below. We have read this article "High-quality chromosome-scale assembly of the walnut (Juglans regia L.) reference genome". This article mentions in the Availability of Supporting Data and Materials section that all raw data can be found in the NCBI BioProject database PRJNA291087. Therefore, we searched for PRJNA291087 in NCBI (https://www.ncbi.nlm.nih.gov/bioproject/PRJNA291087/ ) and found GCA_001411555.2 (Figure 3). Next, we went to the Walnut 2.0 reference genome interface ( https://www.ncbi.nlm.nih.gov/datasets/genome/GCA_001411555.2/ ). We found the annotation results of the Walnut 2.0 genome on this website (GCF001411555.2-Walnut 2.0_genomic. gff. gz). In this annotation result, all gene IDs are represented in the format LOC118344036 (Figure 4). So, the gene ID format presented in Supplementary Table S9 is consistent with the annotation results. In order to find the source of the Jr01_00010 gene ID format you mentioned, we read the article mentioned above. We found the Jr01_00010 format you mentioned in Table S10, Table S12, and genome version 2. As mentioned in Table S10, the Jr01_00010 format is v2 ID, and XP_018814383 is the ID submitted to NCBI. The LOC118348327 format in our table can correspond to the XP_018814383 format box (Figure 4). What we can confirm is that the annotation result for Walnut 2.0 (GCF001411555.2) in NCBI does not include the format Jr01_00010. In order to ensure that readers can correctly search for the reference genome used in this article, we have added the website address in the text ( https://www.ncbi.nlm.nih.gov/datasets/genome/GCF_001411555.2/). And it was clarified that the reference genome number is GCF001411555.2. Thank you again for your question. Our data is indeed genuine and reliable. We hope this answer can dispel your doubts about our data. If you have any further questions, we hope you can contact us promptly (lg20170404@bjfu.edu.cn).
Figure 1

Figure 2

Figure 3

Figure 4

Round 4
Reviewer 3 Report
Comments and Suggestions for Authors
It can be accepted for publication in its current form.